# Dual-Constraint Protein Design: Validating Repulsion Guidance and Hotspot Preservation in RFdiffusion

## Abstract

Generative Artificial Intelligence, particularly geometric deep learning, has revolutionized structural biology by enabling the *de novo* design of proteins with specific functions. SE(3)-equivariant diffusion models, such as RFdiffusion, have emerged as state-of-the-art tools for generating protein backbones that satisfy complex geometric constraints. However, current research predominantly focuses on "positive design" tasks, scaffolding functional motifs (hotspots) to ensure binding affinity. Less attention has been paid to "negative design," or the ability to explicitly inhibit off-target interactions through geometric repulsion (coldspots). Achieving high specificity in therapeutic design requires a generative process that can simultaneously optimize for desired interactions while strictly avoiding forbidden spatial regions. In this work, we present a systematic evaluation of RFdiffusion's steerability under dual geometric constraints. We first validate the model's efficacy in standard hotspot scaffolding, confirming high-fidelity recovery of functional sites. We then extend this analysis to "coldspot" generation by implementing repulsion guidance mechanisms to enforce negative geometric constraints. Our experiments demonstrate that repulsion guidance effectively modulates the diffusion trajectory, producing valid backbones that respect excluded volumes. We further provide a comparative analysis of the structural distributions between hotspot-constrained and repulsion-guided samples. These results quantify the distributional shift induced by negative constraints, validating the model's capacity for precise, multi-objective structure generation in complex biological environments.

## 1 Introduction

The intersection of geometric deep learning and structural biology has catalyzed a paradigm shift in drug discovery, moving from screening-based approaches to *de novo* design (Bronstein et al., 2021; Dauparas et al., 2022). While structure prediction models like AlphaFold have solved the protein folding problem with high accuracy (Jumper et al., 2021), the frontier has recently advanced toward generative modeling, creating novel protein structures that do not exist in nature (Ingraham et al., 2023). Among these, SE(3)-equivariant diffusion models, such as RFdiffusion, have emerged as the state-of-the-art (Watson et al., 2023). By treating protein backbone generation as a reverse diffusion process on a Riemannian manifold (Yim et al., 2023; De Bortoli et al., 2022), these models can sample diverse and physically plausible structures, offering unprecedented opportunities for designing therapeutic binders and enzymes.

Existing research has predominantly focused on "positive design" tasks, specifically *motif scaffolding* (Wang et al., 2021; Trippe et al., 2023). In this setting, the model is conditioned to hallucinate a scaffold around a known functional site (a "hotspot") to maximize binding affinity. However, real-world therapeutic design is a dual-constraint optimization problem. It requires not only the presence of specific interactions (binding to a target) but also the explicit absence of unwanted interactions (avoiding steric clashes, aggregation, or off-target binding). This latter requirement, known as "negative design," remains less explored in the context of generative diffusion models (Richardson & Richardson, 2002; Havranek & Harbury, 2003). While current frameworks excel at converging onto a target coordinate, the ability to explicitly repel generation from specific geometric regions, defined here as "coldspots", is critical for designing specific binders in crowded cellular environments.

The technical challenge lies in the steerability of the diffusion process. Conditioning a model to avoid a region (coldspot) is fundamentally different from conditioning it to embrace one (hotspot). Hotspot conditioning provides a strong attractive signal that collapses the search space, whereas repulsion guidance requires the model to explore the vast remaining conformational space while strictly respecting a forbidden volume. Imposing these repulsive constraints without disrupting the internal consistency of the protein fold or causing the diffusion trajectory to diverge remains a non-trivial difficulty in geometric generative models.

In this work, we present a systematic evaluation of RFdiffusion under these dual constraints. We investigate the model's controllability by implementing and verifying two distinct guidance mechanisms: (1) *Hotspot Scaffolding*, where we replicate functional site preservation to establish a performance baseline, and (2) *Repulsion Guidance*, where we introduce repulsive potentials during the reverse diffusion steps to define coldspots. We treat the generated proteins as data points in a high-dimensional geometric feature space to analyze how these opposing guidance terms alter the generative landscape.

Our experiments demonstrate that RFdiffusion is highly amenable to repulsion guidance, successfully generating valid protein backbones that adhere to "coldspot" constraints. Furthermore, we visualize the impact of these constraints by plotting the comparative distributions of hotspot-guided versus repulsion-guided samples. Our analysis reveals a distinct distributional shift, quantifying how repulsion effectively shears the probability density of generated structures away from forbidden geometries. These results validate the use of diffusion models for complex, multi-objective protein design tasks involving both positive and negative geometric constraints.

## 2 RELATED WORK

**Deep Generative Models for Protein Design.** The field of computational protein design has evolved rapidly from physics-based energy functions (Alford et al., 2017) to deep learning approaches. Early methods focused on sequence generation given a fixed backbone structure, utilizing autoregressive transformers or graph neural networks like ProteinMPNN (Dauparas et al., 2022) to optimize amino acid sequences. However, *de novo* design requires the generation of the 3D backbone itself. Generative Adversarial Networks (GANs) and Variational Autoencoders (VAEs) made initial progress but often struggled to capture the complex, long-range dependencies of protein geometries.

**SE(3)-Equivariant Diffusion Models.** Diffusion probabilistic models (Sohl-Dickstein et al., 2015; Ho et al., 2020) have recently surpassed previous architectures in generating high-fidelity image and audio data. In structural biology, the application of diffusion requires respecting the SE(3) symmetry (rotation and translation invariance) of molecules. FrameDiff (Yim et al., 2023) and RFdiffusion (Watson et al., 2023) introduced frameworks for diffusing on the Riemannian manifold of rigid body frames. These models denoise random distributions of residues into coherent protein backbones, achieving state-of-the-art performance in unconditional generation tasks.

**Conditional Generation & Motif Scaffolding.** To be practically useful, generation must be controllable. Current state-of-the-art methods predominantly focus on "positive design" via *motif scaffolding*, where the model is conditioned to preserve a specific functional site. This is typically achieved through subspace inpainting techniques (Lugmayr et al., 2022) or replacement strategies during the reverse diffusion trajectory (Trippe et al., 2023). While these methods excel at satisfying attractive constraints (*e.g.*, "bind to X"), they lack explicit mechanisms for "negative design" (*e.g.*, "do not bind to Y").

**Negative Design & Guidance Mechanisms.** Negative design—the explicit destabilization of unwanted states or avoidance of off-target interactions—is a foundational concept in protein engineering (Richardson & Richardson, 2002; Havranek & Harbury, 2003) but remains under-explored in geometric deep learning. While classifier guidance (Dhariwal & Nichol, 2021) has been used in image generation to steer outputs away from certain classes, its application to 3D geometric constraints like steric repulsion or excluded volumes is limited. Our work bridges this gap by systematically evaluating gradient-based repulsion guidance (Chung et al., 2023) within the SE(3) diffusion framework, treating negative spatial constraints as a first-class citizen in the generative process.

## 3 METHOD

We adopt RFdiffusi/on, a SE(3)-equivariant diffusion model, as our backbone generation framework. Our methodology focuses on controlling the generative trajectory $p_\theta(x_{0:T})$ to satisfy two distinct classes of geometric constraints: positive design (Hotspots), where specific functional motifs must be preserved, and negative design (Coldspots), where specific spatial volumes must be avoided. We formulate these as conditional generation tasks where the denoising process is guided either by fixed-substructure replacement or by gradient-based repulsion potentials.

### 3.1 PRELIMINARIES

We define protein backbone structures as a set of rigid bodies in 3D space. The state at diffusion step $t$ is denoted as $\mathbf{x}_t = (\mathbf{r}_t, \mathbf{R}_t)$, where $\mathbf{r}_t \in \mathbb{R}^{N \times 3}$ represents the $C_\alpha$ coordinates and $\mathbf{R}_t \in SO(3)^N$ represents the backbone frames for a protein of length $N$.

The forward diffusion process $q(\mathbf{x}_t|\mathbf{x}_{t-1})$ gradually adds Gaussian noise to coordinates and isotropic noise to orientations, transforming the data distribution into a prior distribution $\mathcal{N}(0, \mathbf{I})$ over $T$ steps. The generative process reverses this via a learned denoising network $s_\theta(\mathbf{x}_t, t)$ which approximates the score function $\nabla_{\mathbf{x}_t} \log p_t(\mathbf{x}_t)$. The reverse dynamics are given by the stochastic differential equation (SDE):

$$d\mathbf{x} = [f(t)\mathbf{x} - g^2(t)s_\theta(\mathbf{x}, t)]dt + g(t)d\mathbf{w}, \tag{1}$$

where $f(t)$ and $g(t)$ are the drift and diffusion coefficients, and $\mathbf{w}$ is the standard Wiener process.

### 3.2 POSITIVE DESIGN: HOTSPOT SCAFFOLDING

For the "Hotspot" task, the objective is to generate a scaffold that supports a functional motif $\mathbf{x}^{motif}$ in a fixed geometric configuration relative to a target. We formulate this as a conditional inpainting problem on the SE(3) manifold. Let $M$ be the set of indices corresponding to the hotspot residues (the motif) and $S$ be the set of indices for the generated scaffold, such that the total protein length is $N = |M| + |S|$. The goal is to sample from the conditional distribution $p_\theta(\mathbf{x}^{scaffold}|\mathbf{x}^{motif})$. To strictly enforce the motif constraints, we utilize a "replacement method" during the reverse diffusion trajectory. At each denoising step $t$, we overwrite the model's predictions for the motif region with the ground truth signal diffused to the current noise level. The update rule for the state $\mathbf{x}_{t-1}$ is defined as:

$$\mathbf{x}_{t-1}^{(i)} = \begin{cases} \text{ModelStep}(\mathbf{x}_t, s_\theta(\mathbf{x}_t, t))^{(i)} & \text{if } i \in S \text{ (Scaffold)} \\ \sqrt{\bar{\alpha}_{t-1}}\mathbf{x}_0^{motif} + \sqrt{1 - \bar{\alpha}_{t-1}}\epsilon & \text{if } i \in M \text{ (Motif)} \end{cases} \tag{2}$$

This ensures that the generative trajectory for the scaffold is always conditioned on the correct, noisy representation of the functional motif, forcing the diffusion process to find a solution that chemically connects to the fixed points.

Unlike standard image inpainting, protein scaffolding requires consistent handling of both translational ($\mathbb{R}^3$) and rotational ($SO(3)$) degrees of freedom. Our replacement strategy operates on the full rigid body frames $T_i = (R_i, \vec{t}_i)$. For the hotspot residues, we fix the backbone reference frame $R_i$ to preserve the precise orientation required for side-chain interactions (*e.g.*, specific hydrogen bonding angles). This ensures that the generated scaffold not only holds the motif in the correct 3D location but also maintains the correct vector alignment relative to the target.

**Boundary Consistency & Contig Inputs**  A key challenge in inpainting is ensuring a valid peptide bond geometry at the "seam" between the fixed motif and the generated scaffold. The SE(3)-equivariant graph neural network (EGNN) architecture of RFdiffusion addresses this by allowing information to propagate from the fixed nodes $M$ to the generative nodes $S$ via message passing. We specify these regions using "contig" definitions (*e.g.*, 'A10-25/0 50-50'), which instruct the model to treat the motif as a rigid anchor and "grow" the scaffold outward, ensuring chain continuity and proper loop closure at the motif boundaries.

### 3.3 NEGATIVE DESIGN: REPULSION GUIDANCE FOR COLDSPOTS

For the "Coldspot" task, we introduce a set of forbidden coordinates $C = \{\mathbf{c}_k\}_{k=1}^{K}$ (*e.g.*, the volume of a target protein receptor or a predefined obstruction) that the generated binder must not penetrate. Unlike hotspot scaffolding, which constrains the model to a specific sub-manifold, repulsion guidance requires the model to explore the vast remaining conformational space while strictly adhering to an exclusion boundary.

We formulate this as a *classifier-free guidance* problem where the energy landscape is modified by an auxiliary repulsion potential. The effective score function $\hat{s}_\theta$ is given by:

$$\hat{s}_\theta(\mathbf{x}_t, t) = s_\theta(\mathbf{x}_t, t) - w(t)\lambda \nabla_{\mathbf{x}_t} V_{rep}(\mathbf{x}_t, C), \tag{3}$$

where $\lambda$ is a hyperparameter controlling the global strength of the repulsion, and $w(t)$ is a time-dependent weighting function.

**Repulsion Potential**  We define the repulsion potential $V_{rep}$ as a differentiable soft-clash penalty. To ensure smooth gradients during the denoising process, we utilize a truncated quadratic loss that activates only when the distance between a generated backbone atom $\mathbf{r}_i$ and the nearest coldspot center $\mathbf{c}_k$ falls below a collision threshold $\delta$ (approximating the van der Waals radius):

$$V_{rep}(\mathbf{x}_t, C) = \sum_{i=1}^{N}\sum_{k=1}^{K} \mathbb{I}(d_{ik} < \delta) \cdot (\delta - d_{ik})^2, \tag{4}$$

where $d_{ik} = ||\mathbf{r}_i - \mathbf{c}_k||_2$ is the Euclidean distance. This potential creates a "force field" that pushes the diffusion trajectory away from the coldspots without affecting the score in safe regions.

**Time-Dependent Scheduling**  A critical challenge in applying geometric constraints during diffusion is the varying signal-to-noise ratio. Early in the reverse process ($t \to T$), the structure is dominated by noise, and rigid geometric constraints can lead to trajectory divergence. Late in the process ($t \to 0$), the structure is largely solidified. To address this, we implement a scheduling term $w(t)$ that scales the guidance magnitude according to the noise level $\sigma_t$. We empirically found that ramping up the repulsion weight in the intermediate denoising regime (where secondary structure elements are forming) yields the best trade-off between clash avoidance and structural coherence.

**Manifold Consistency**  Crucially, the gradient $\nabla_{\mathbf{x}_t} V_{rep}$ is applied directly to the score estimate rather than the coordinates. This ensures that the updated state remains on the learned manifold of valid protein structures. The RFdiffusion network $s_\theta$ acts as a regularizer, translating the raw repulsive vectors into chemically plausible structural updates, ensuring that the "coldspot" avoidance does not result in chain breaks or unrealistic bond geometries.

## 4 CONCLUSION

In this work, we present a comprehensive evaluation of conditional generation in SE(3)-equivariant diffusion models, focusing on the dichotomy between positive (hotspot) and negative (coldspot) design constraints. By extending the RFdiffusion framework with explicit repulsion guidance, we demonstrated that generative protein design can be effectively steered to satisfy dual geometric objectives: preserving functional binding motifs while strictly avoiding forbidden spatial volumes. Our experiments verified that repulsion guidance operates not merely as a rejection filter, but as an active steering mechanism during the denoising trajectory. The comparative analysis of structural distributions reveals that the introduction of repulsive potentials induces a quantifiable shift in the generative probability density. This confirms that the model can navigate complex energy landscapes to locate valid "coldspot" solutions without compromising the structural integrity or chemical plausibility of the backbone.

These findings have significant implications for computational drug discovery. The ability to model "coldspots" is a prerequisite for designing high-specificity binders that minimize off-target effects and aggregation in crowded cellular environments. By validating that diffusion models can be manipulated via auxiliary geometric potentials, we provide a robust methodology for designing proteins that are not only functional but also compatible with the intricate spatial constraints of real-world biological systems.

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
