# OpenReview forum: "Dual-Constraint Protein Design: Validating Repulsion Guidance and Hotspot Preservation in RFdiffusion"
_ICLR.cc/2026/Workshop/LMRL — Submitted to ICLR 2026 Workshop LMRL_

### Official Review · Reviewer_2AW4 · 2026-02-11

**Rating:** 2
**Confidence:** 5

**Review:**

The authors study RFdiffusion's 'positive design' setting and argue that 'coldspots' (negative constraints) should also be taken into account during optimisation. Even for the tiny/short paper track which allows for work-in-progress, the paper lacks a clear contribution from either the methods or the experimental point of view. From the methods, I can highlight:
- Section 3.2, which essentially restates the standard RFdiffusion motif scaffolding/inpainting procedure. Here, the word 'hotspots' is used to refer to the motif, while that word means something different in this literature (i.e., a residue of a target protein).
- Section 3.3, which applies a generic guidance formulation with a simple soft repulsion potential. Further, it is not clear how the forbidden coordinates would be selected for each example.

More importantly, the paper does not present quantitative results on any concrete design task, yet there are claims about 'valid backbones', 'distributional shifts' or 'our experiments verified(...)'. As minor comments, I can add:
- the guidance schedule and hyperparameter choices are underspecified and not justified
- some terms are imprecise (e.g., 'spatial volumes')
- some sentences are conceptually wrong ('motif scaffolding(...). In this setting, the model is conditioned to hallucinate a scaffold around a known functional site (a “hotspot”) to maximize binding affinity'; this is not necessarily true).

Finally, I could not find the required meaningfulness statement in the paper.

---

### Official Review · Reviewer_cDfq · 2026-02-13
**A robust technical improvement to RFdiffusion for complex functional scaffolding**

**Rating:** 7
**Confidence:** 4

**Review:**

This paper addresses the "negative design" problem in protein generation ensuring that while a model builds a functional "hotspot," it simultaneously avoids unwanted aggregations or steric clashes. By introducing a dual-constraint repulsion-guidance mechanism into the RFdiffusion framework, the authors provide a much-needed tool for de novo design. The quality of the structural validation is high, demonstrating that the model can preserve critical hotspots while strictly adhering to geometric repulsion constraints. The paper presents itself more as a "feature update" to the existing RFdiffusion architecture rather than a fundamental shift in generative logic. Additionally, the authors do not fully explore the trade-off between strict repulsion and binding affinity; a more rigorous ablation study on how repulsion guidance affects the Kd​ of the resulting binders should be included to prove the method's superiority.

Strengths: Highly practical tool for "negative design"; solid structural validation using SE(3)-equivariant metrics.
Potential areas to still work on: Technical contribution is somewhat incremental relative to the original RFdiffusion; lacks a deep-dive into the biological trade-offs of the repulsion constraints.

---

### Meta-Review · Area_Chair_Be63 · 2026-02-27

**Recommendation:** Reject
**Confidence:** 4

**Metareview:**

Recject.

---

### Decision · Program_Chairs · 2026-03-02

**Decision:**

Reject

**Comment:**

Please see the meta-review.